# The Effects of High-Intensity, Short-Duration and Low-Intensity, Long-Duration Hamstrings Static Stretching on Contralateral Limb Performance

**DOI:** 10.3390/sports12090257

**Published:** 2024-09-18

**Authors:** Emily J. Philpott, Mohammadmahdi Bahrami, Mahta Sardroodian, David G. Behm

**Affiliations:** School of Human Kinetics and Recreation, Memorial University of Newfoundland, St. John’s, NL A1C 5S7, Canada; ejphilpott@mun.ca (E.J.P.); mohammadmahb@mun.ca (M.B.); msardroodian@mun.ca (M.S.)

**Keywords:** range of motion, maximal voluntary isometric contraction, muscle activation, stretch tolerance, flexibility

## Abstract

Introduction: Increases in contralateral range of motion (ROM) have been shown following acute high-intensity and high-duration static stretching (SS) with no significant change in contralateral force, power, and muscle activation. There are currently no studies comparing the effects of a high-intensity, short-duration (HISD) or low-intensity, long-duration (LILD) SS on contralateral performance. Purpose: The aim of this study was to examine how HISD and LILD SS of the dominant leg hamstrings influence contralateral limb performance. Methods: Sixteen trained participants (eight females, eight males) completed three SS interventions of the dominant leg hamstrings; (1) HISD (6 × 10 s at maximal point of discomfort), (2) LILD (6 × 30 s at initial point of discomfort), and (3) control. Dominant and non-dominant ROM, maximal voluntary isometric contraction (MVIC) forces, muscle activation (electromyography (EMG)), and unilateral CMJ and DJ heights were recorded pre-test and 1 min post-test. Results: There were no significant contralateral ROM or performance changes. Following the HISD condition, the post-test ROM for the stretched leg (110.6 ± 12.6°) exceeded the pre-test (106.0 ± 9.0°) by a small magnitude effect of 4.2% (*p* = 0.008, d = 0.42). With LILD, the stretched leg post-test (112.2 ± 16.5°) exceeded (2.6%, *p* = 0.06, d = 0.18) the pre-test ROM (109.3 ± 16.2°) by a non-significant, trivial magnitude. There were large magnitude impairments, evidenced by main effects for testing time for force, instantaneous strength, and associated EMG. A significant ROM interaction (*p* = 0.02) showed that with LILD, the stretched leg significantly (*p* = 0.05) exceeded the contralateral leg by 13.4% post-test. Conclusions: The results showing no significant increase in contralateral ROM with either HISD or LILD SS, suggesting the interventions may not have been effective in promoting crossover effects.

## 1. Introduction

Static stretching (SS) is the most prevalent method of stretching in fitness, sport, and rehabilitation [1,2,3], where an individual holds a stationary stretching position, placing tension on the muscle tendon unit often to the maximum range of motion (ROM) or point of discomfort (POD). The optimal SS prescription regarding the dose–response relationship between SS duration and intensity on ROM and maximal muscle performance has been heavily debated for many years. In the current literature, it is often suggested that longer durations of SS (>60 s per muscle group) prior to activity is more likely to cause sport or work muscle performance impairments (i.e., force, power, and vertical jump height) than shorter duration SS (<60 s per muscle group) [4,5,6,7,8,9]. In some studies, acute bouts of higher intensity SS have also been suggested to result in greater ROM increases than lower-intensity SS, although low-intensity SS still demonstrates increases in ROM [3,10,11,12]. However, recent meta-analytical reviews provide evidence that acute or chronic SS intensity may not moderate the effects of muscle strength, power [13], or ROM [5,14] and there is no association between high-duration chronic SS training and increased ROM [14]. Thus, the recommended SS prescription remains unclear.

Several original studies [15,16,17,18,19,20,21] have reported an increase in crossover (non-local or contralateral limb) ROM following an acute bout of SS. A meta-analysis by Behm et al. [22] reported that 240 s of SS exhibited large magnitude (d = 1.24) increases in non-local ROM compared to moderate magnitude (d = 0.72) improvements with shorter (<120 s) SS durations. As the non-local or contralateral limb is not physically stretched in these scenarios, it is postulated that the increase in ROM cannot be attributed to morphological mechanisms such as increased musculotendinous unit compliance [16,18,19,23]. Increased stretch tolerance is a commonly suggested mechanism underlying an increase in ROM of the stretched limb [5,6,24,25,26] as well as with non-stretched contralateral and non-local muscles and joints [16,18,19,21,22,23,27].

Several studies exhibiting these acute non-local ROM changes have not reported a decrease in contralateral limb force, power [19,28], or muscle activation [16,18]. For example, Hadjizadeh Anvar et al. [16] showed significant increases in contralateral ROM following 6 × 45 s SS of the dominant leg plantar flexors to the point of discomfort with no significant decrease in contralateral force. Similarly, Behm et al. [18] showed significant increases in stretched and non-stretched contralateral leg ROM following 8 × 30 s SS of the dominant leg quadriceps and hamstrings with no significant changes in maximal voluntary isometric contractions (MVIC) or muscle activation, further emphasizing an increase in stretch tolerance as a mechanism for this increase in ROM. Global pain/stretch modulatory systems such as diffuse noxious inhibitory control and gate control theory are suggested to contribute to an increase in stretch tolerance [16,22,23,29] causing a suppressed transmission or analgesic effect on pain throughout the body and therefore an increased ability to withstand discomfort and push through a greater ROM [16,22,23,29]. Several studies examining the effects of acute SS on contralateral limb performance typically administer high-intensity (to the maximum point of discomfort), high-duration (>60 s) SS [16,18,19]. There are no studies investigating the effects of high-intensity, short-duration (HISD) or low-intensity, long-duration (LILD) SS on contralateral limb performance.

Thus, the purpose of the present study was to examine the acute effects of HISD and LILD SS of the dominant leg hamstrings on contralateral limb performance. It was hypothesized that greater contralateral limb ROM would be seen following the unilateral HISD intervention, and both the unilateral HISD and LILD interventions would show no significant change in contralateral limb performance (isometric force, muscle activation, countermovement jump (CMJ), and drop jump (DJ) height).

## 2. Methods

### 2.1. Participants

A statistical power analysis was completed (G*Power 3.1.9.7) based on similar studies [16,30] with large magnitude partial eta-squared (η_p_^2^) effect sizes for force and EMG measures ranging from 0.16 to 0.29 [16] and Cohen’s d effect sizes of 0.7 to 2.85 [30]. A sample size of 16 participants was determined to be needed to achieve an α value of *p* < 0.05 with a power of 0.8. Convenience sampling was used to recruit 16 apparently healthy, trained males (*n =* 8) and females (*n =* 8). All participants had a minimum of 1 year of resistance training experience, resistance trained at least 3× per week, and regularly stretched either prior to or following their resistance training workouts. All participants did not have lower limb injuries that currently present symptoms nor undergone a lower limb surgical procedure in the past. Participants had a mean age of 21.5 ± 1.41 years and resistance training experience of 4.4 ± 2.5 years. Mean height of 179.68 ± 5.49 cm for males and 164.81 ± 3.67 cm for females and mean weight of 88.4 ± 11.7 kg for males and 69.2 ± 16.7 kg for females were recorded. Participants were asked to maintain their regular training routine and to refrain from participating in unusually strenuous activities and consuming caffeine and recreational drugs or alcohol within 24 h prior to each data collection session. Each participant was extensively informed of the procedure and signed an informed consent form prior to testing. Ethical approval was granted by the institution’s Interdisciplinary Committee on Ethics in Human Research (ICEHR # 20241205-HK). Experimental procedures were conducted in accordance with the 2013 Declaration of Helsinki.

### 2.2. Experimental Design

The study consisted of a familiarization session and three testing sessions of approximately 30 min each, each separated by a minimum of 24 h. During the familiarization session, the roles of the participants were verbally explained, and they read and signed the consent form. Anthropometric measures were recorded, and the intervention and testing procedures were explained, demonstrated, and practiced. At the beginning of each testing session, participants completed a 5 min aerobic warm-up (Monark^®^ cycle ergometer, Monark, Stockholm, Sweden) at 70 RPM and 1 kilopond resistance. Participants were asked to identify their dominant lower limb, defined by the lower limb used in a kicking task. SS of the dominant leg hamstrings was administered passively by a researcher as an intervention. Participants completed either (1) HISD (6 × 10 s at maximal POD), (2) LILD (6 × 30 s at initial POD), or (3) passive control during each session. Participants were allocated a 10 s recovery period between each SS. The intervention implemented during each session was randomized. During pre- and post-test measures, measurements of dominant and non-dominant lower limbs were randomized, and the order of measurement was also randomized. Both pre- and post-test, participants completed 2–3 trials of hamstring ROM, knee flexion MVIC, unilateral CMJ, and unilateral DJ. A third trial was administered if the second trial was 5% greater than the first trial.

### 2.3. Pre- and Post-Test Measures

#### 2.3.1. Hamstrings Range of Motion (ROM)

Participants were asked to lie supine on a padded table with their legs extended and arms by their side. The participants’ greater trochanter of the femur and lateral malleolus was then palpated, and a digital goniometer (EasyAngle^®^, Meloq, Stockholm, Sweden) was then placed at those points. The same researcher for each ROM measure then flexed the participant’s hip, passively ensuring the stretched leg’s knee remained straight (extended) and the non-stretched leg remained on the table. The hip was flexed until the participant verbally communicated to the researcher that the maximal point of discomfort was reached. Two ROM measurements were completed with a third measurement if the second measurement was 5% greater than the previous measurement. There was a 10 s rest period between each ROM trial. This procedure was repeated for both the dominant and non-dominant lower limbs.

#### 2.3.2. Knee Flexion Maximal Voluntary Isometric Contraction (MVIC) Force

Participants were seated upright in a seat with the back against a backrest and the hips at approximately 90°. Their hips and trunk were secured to the seat with an adjustable strap. A cuff was placed around the ankle and connected to a chain, which was attached to a strain gauge (Omega Engineering Inc., LCCA 250, Don Mills, ON, Canada) in front of the seated participant. The knee was positioned at 120° of knee flexion and the ankle at 90° (Figure 1). As a warm-up, participants completed two knee flexion isometric contractions at approximately 50% of their perceived maximal contraction and one contraction at approximately 90% of their subjective maximal contraction. Each contraction was held for 3 s.

Pre-test knee flexion MVICs were performed for two trials with 1 min rest between trials. A third trial was performed if the second MVIC was 5% greater than the first MVIC. This procedure was repeated for both the dominant and non-dominant lower limbs. Post-test knee flexion MVICs were performed in the same manner without a prior warm-up. MVIC measures were acquired by a strain gauge and digitally transferred to a data acquisition software 5.0 (BIOPAC^®^ Systems Inc., Holliston, MA, USA). Differential voltage (±0.03% linearity and 3 mv/V) from the strain gauges, sampled at a rate of 2000 Hz, were calibrated (to Newtons), amplified (×1000), digitally converted (Biopac Systems Inc., DA 100 and analog to digital converter MP100WSW; Holliston, MA, USA), and monitored on a computer. A commercial software program (AcqKnowledge III, Biopac Systems Inc., Holliston, MA, USA) was used to analyze the digitally converted analog data. The highest peak force of the two to three MVIC trials were used for analysis (AcqKnowledge, BIOPAC^®^ Systems Inc.). Instantaneous strength (peak force exerted in the first 100 ms) was also extracted from the contraction with the highest force. The onset of instantaneous strength was determined by visual inspection of the force output where the force line initially deviated from the baseline without returning to the baseline and continued to escalate.

#### 2.3.3. Hamstrings Activation (Electromyography: EMG)

The participants’ skin in the area midway between the gluteal fold and popliteal fossa of the knee was shaved of hair, abraded using an abrasive gel, and cleaned using an alcohol swab. The distance between the gluteal fold and popliteal fossa was measured using a soft tape measure, and the midpoint of the biceps femoris muscle belly was marked using a skin-safe marker. Surface EMG electrodes (Covidien Kendall™, Cardinal Health, Dublin, OH, USA) were placed on the skin of the indicated area as reported by SENIAM recommendations. The mean amplitude of the root mean square (RMS) EMG activity was recorded 0.5 s before and after the peak force of each knee flexion MVIC. All EMG signals were monitored (Biopac System Inc., DA 100: analog–digital converter MP150WSW; Holliston, MA, USA) and recorded with a sampling rate of 2000 Hz using AcqKnowledge III, Biopac System Inc. software. EMG activity was filtered with a Blackman −61 dB band-pass filter between 10 and 500 Hz, amplified (bi-polar differential amplifier, input impedance = 2 MΩ, common-mode rejection ratio > 110 dB min (50/60 Hz), gain × 1000, noise > 5 µV), and analog-to-digitally converted (12 bit) and stored on a personal computer for further analysis.

#### 2.3.4. Unilateral Countermovement Jump (CMJ) and Drop Jump (DJ) Height

Unilateral CMJ and DJ heights were recorded using a ChronoJump Boscosystem^®^ (Barcelona, Spain) linear encoder software. A belt was strapped around the participants’ waist with the lower edge in line with the iliac crest, and the linear encoder was attached to the belt at the participants’ left hip. The participants were instructed to stand unilaterally with hands on the hips above the belt (akimbo). For the CMJ, participants were verbally instructed by the researcher to perform a brief knee flexion to a depth of their personal preference and then jump as high as possible landing on the same foot. Participants initiated the jump at their personal discretion (i.e., when they felt ready). Knee flexion times and depths were self-selected by the participants; however, participants were encouraged to minimize the time and depth. Hands remained on the hips throughout the jump. Two trials were completed for both dominant and non-dominant legs.

During DJ trials, the belt and linear encoder were arranged in the same manner. DJ height trials commenced with participants standing on a platform 20 cm high and stepping off (no prior vertical jump component) onto the tested leg. Participants were instructed to rebound as quickly as possible when the foot hits the ground and to minimize knee flexion time and depth during the rebound. Two trials were completed for both dominant and non-dominant legs.

### 2.4. Control Session

The control session was executed in the same manner as the HISD and LILD sessions with the exception of the dominant leg SS intervention. Following pre-test measures, participants were instructed to lie supine on a padded table with arms by their sides and legs straight. Participants remained in this position for 255 s, which is equivalent to the duration of the LILD SS intervention. Post-test measures were recorded 1 min following the intervention.

### 2.5. Statistical Analysis

Statistical analyses were calculated using SPSS software (Version 28.0, SPSS, Inc., Chicago, IL, USA). Kolmogorov–Smirnov tests of normality were conducted for all dependent variables. An α value of *p* < 0.05 was considered statistically significant. If the assumption of sphericity was violated, the Greenhouse−Geiser correction was employed. Sex differences for the stretched and non-stretched legs were considered with a repeated measure 3-way ANOVA (2 sexes × 3 interventions × 2 test times). However, as there were no significant sex differences detected, the data were then re-analyzed using a repeated measure 3-way ANOVA (2 conditions × 3 interventions × 2 test times). This included 2 conditions (stretched leg versus non-stretched leg), 3 interventions (HISD, versus LILD versus control), and 2 test times (pre-test versus post-test). Bonferroni post hoc tests were conducted to detect significant main effect differences whereas for significant interactions, Bonferroni post hoc *t*-tests corrected for multiple comparisons (α-value divided by the number of analyses on the dependent variable) were conducted to determine differences between values. Partial eta-squared (η_p_^2^) values are reported for main effects and overall interactions representing small (0.01 ≤ η_p_^2^ < 0.06), medium (0.06 ≤ η_p_^2^ < 0.14), and large (η_p_^2^ ≥ 0.14) magnitudes of change (from SPSS-tutorials, 2022). Cohen’s d effect sizes are reported for the specific post hoc interactions with d < 0.2: trivial, 0.2–<0.5: small, 0.5–<0.8: moderate, and ≥0.8: large magnitude difference [31]. Significance was established at *p* ≤ 0.05.

## 3. Results

### 3.1. Range of Motion

A significant main effect for the leg (F_(1,14)_ =4.71, *p* = 0.048, eta^2^: 0.25) revealed that the stretched leg demonstrated a 1.9% greater ROM (109.12 ± 12.5°) than the contralateral non-stretched leg (107.14 ± 13.36°). A significant main effect for testing time (F_(1,14)_ =5.52, *p* = 0.034, eta^2^: 0.28) showed that ROM increased 1.7% (d = 0.14) from pre- (107.2 ± 12.0°) to post-test (109.05 ± 13.6°). There was a near significant leg × stretch-intensity duration × testing time interaction (F_(2,28)_ =2.74, *p* = 0.08, eta^2^: 0.16). Following the HISD condition, the post-test ROM for the stretched leg (110.6 ± 12.6°) exceeded the pre-test (106.0 ± 9.0°) by 4.2% (*p* = 0.008, d = 0.42). Similarly, with LILD, the stretched leg post-test (112.2 ± 16.5°) also exceeded (*p* = 0.06, d = 0.18) the pre-test ROM (109.3 ± 16.2°) by 2.6% (Table 1).

### 3.2. MVIC Force and Instantaneous Strength

With peak force, there was a significant main effect for testing time (F_(1,13)_ =40.13, *p* < 0.001, eta^2^: 0.75) with an overall 3.7% decrease from pre- (343.1 ± 108.4 N) to post-testing (330.6 ± 105.7 N). With instantaneous strength, there was also a significant main effect for testing time (F_(1,13)_ =10.14, *p* = 0.007, eta^2^: 0.44) with an overall 10.9% decrease from pre- (114.8 ± 80.2 N) to post-testing (102.4 ± 77.7 N). Similarly, the EMG associated with instantaneous strength (EMG muscle activation measured over the first 100 ms of force production) also revealed a significant, large magnitude, main effect for testing time (F_(1,12)_ =18.31, *p* = 0.001, eta^2^: 0.604) with an overall 10.2% decrease from pre- (0.059 ± 0.036 mV) to post-testing (0.053 ± 0.036 mV).

### 3.3. Unilateral Countermovement (CMJ) and Drop Jump (DJ) Height

There were no significant main effects or interactions for CMJ height. There was a non-significant main effect (F_(1,13)_ =3.21, *p* = 0.09, eta^2^: 0.19) for legs with DJ height with the stretched leg (10.7 ± 4.2 cm) demonstrating a 9.4% greater jump height than the contralateral non-stretched leg (9.7 ± 4.0 cm). A significant leg × stretching intensity × testing time interaction (F_(2,26)_ =4.52, *p* = 0.02, eta^2^: 0.26) for DJ showed that with LILD, the stretched leg (10.5 ± 3.7 cm) significantly (*p* = 0.05, d = 0.38) exceeded the contralateral leg (9.1 ± 3.8 cm) by a small magnitude 13.4% at post-test (Table 1).

## 4. Discussion

The major findings of this study were (1) unilateral stretching did not induce contralateral effects; (2) no significant differences with the stretched leg ROM increases between HISD and LILD SS; (3) overall, SS induced decreases in peak force, instantaneous strength and EMG; and (4) LILD provided greater increases with the stretched leg DJ heights overall.

An increase in contralateral limb ROM is fairly consistent within the current literature [16,18,19,21,23,27]. Behm et al.’s [22] meta-analysis based on 11 studies (14 independent measures) reported moderate magnitude enhancement of non-local or crossover ROM. However, these findings are not unanimous as not all studies have shown non-local ROM increases with either acute [32] or chronic stretching [33,34]. In contrast to the meta-analytical results [22], the present study demonstrated no significant increase in contralateral limb ROM. The majority of studies that provide evidence of an increase in contralateral limb ROM following SS implement longer duration (>60 s) with high-intensity (maximal point of discomfort) SS which differs from the interventions of the current study (high intensity with shorter duration (HISD) vs. lower intensity with longer duration (LILD)). For example, Hadjizadeh Anvar et al. [16] also implemented 180 s (6 × 45 s) of SS, resulting in a significant increase in contralateral limb ROM, but they had participants stretch to the maximum rather than initial POD as with the LILD session of the current study. Behm et al.’s [22] meta-analysis reported that 240 s of SS demonstrated large magnitude non-local ROM increases compared to moderate magnitude improvements with shorter (<120 s) durations. Although the SS prescriptions in this study are in accord with prior reviews [4,6,7,22,35,36], the lack of increase in contralateral limb ROM may suggest 60 s (6 × 10 s) of SS at the maximum point of discomfort (HISD) is an insufficient dosage to stimulate non-local ROM increases. Alternatively, 180 s of SS (6 × 30 s) at the initial point of discomfort (LILD) may be an insufficient intensity to engage crossover ROM effects.

Increased stretch (pain) tolerance has been widely attributed as a primary mechanism underlying stretched and non-stretched joint ROM increases [16,18,19,24,26,37,38,39,40]. Increased stretch tolerance effects to increase contralateral ROM have been attributed to diffuse noxious inhibitory control and gate control theory of pain due to stimulation of nociceptors from the SS which may suppress the sensation of pain [16,18,41,42]. The lower intensity SS with LILD and the lesser duration of SS with HISD may not have elicited a sufficient stimulus for global pain modulation.

An increase in stretched leg ROM following HISD and LILD SS is partially in accordance with recent literature examining the effects of SS [3,5,9,12,14,36,43,44]. In two very similar studies, participants were subjected to 180 s of SS at 80%, 100%, and 120% intensities for either the hamstrings [12] or quadriceps [3], and both studies reported significant increases in ROM following the 100% and 120% intensities but non-significant ROM increases following the 80% intensity SS. With both conditions in the present study, the stretched leg experienced a significant ROM increase following HISD (60 s total at maximal point of discomfort) SS as well as a non-significant (*p* = 0.06) improvement with LILD (180 s total at initial point of discomfort). Subgroup analyses in recent reviews [13,14,45] reported no significant evidence that SS intensity moderates the effect of ROM. The present findings are consistent with the findings of these reviews demonstrating significant and non-significant (*p* = 0.06) increases in stretched leg ROM following both high- and lower-intensity SS, respectively. Practical applications suggest that stretching to the maximal POD may not be required to increase ROM.

The significant, large magnitude overall (main effects for time) decrease in peak force, instantaneous strength, and muscle activation (EMG) are in accord with prior studies and reviews that warn of performance impairments with prolonged SS (≥60 s) without a full dynamic warm-up [4,5,6,7,8,9,35,46]. A possible mechanism includes a decrease in persistent inward currents (PICs) attenuating the gain of the spinal motoneurons [6,47,48,49,50]. It is possible that desensitization of the muscle spindles may disfacilitate spinal motor neurons, decreasing their discharge frequency and adversely affecting the maximal force production [6,47,48,49]. Morphological changes such as decreases in muscle stiffness (increased compliance) have also been reported [4,5,6,7,35,44,51]. But as this finding was a main effect for time and thus combined data from both limbs, this mechanism could not apply to the non-stretched limb. Mental energy deficits (decreased ability to focus or concentrate after an initial bout of exercise) [52,53,54,55] and increases in the perception of effort after an exercise session [56,57] are other possible alternative mechanisms. As mentioned, since these findings were a main effect for time with no significant interactions, this was an overall effect but not specific to either leg.

While there was no significant change in unilateral CMJ height, there was a significant increase in the stretched leg unilateral DJ height with a significant advantage following the LILD intervention. Similar increases were reported by Caldwell et al. [17] who exhibited a significant increase in stretched leg unilateral DJ height following 120 s hamstrings SS with ground contact time being significantly increased (small magnitude) as well. Decreased musculotendinous stiffness (increased compliance) can reduce the efficient transfer of force adversely affecting force development with a rapid stretch shortening cycle (SSC) [58,59,60]. Alternatively, an increase in MTU compliance is conceivably beneficial for tasks that require slower eccentric contractions or a prolonged transition (contact or amortization phase) during the SSC [61,62,63].

This could be a possible mechanism for the present result as a unilateral DJ is a relatively unfamiliar movement; it is likely that the jump required a longer knee flexion phase and contact period to absorb the force compared to the unilateral CMJ. Therefore, an increased MTU compliance may have increased the ability of the musculotendinous unit to store elastic energy over a greater period, leading to an improved unilateral DJ height [4,36]. Similarly, it is noted that forces exerted over an extended duration of unilateral DJ ground contact time could possibly contribute to an increased unilateral DJ height [17]. Unfortunately, ground contact time was not monitored during DJ measures. The current study seems to be in accordance with several previous studies investigating changes in musculotendinous unit compliance following a single bout of SS, implying that only higher duration SS is associated with increased musculotendinous unit compliance [5,6,37,38,64,65] with lower duration SS having minimal or no effect [37,38,65]. This may explain the significant advantage in stretched leg unilateral DJ height seen following the LILD SS intervention.

## 5. Limitations

Every study has limitations that should be considered when interpreting the results. All studies can benefit from a greater number of participants to strengthen the power of the analysis. A great number of participants might have provided greater statistical power to possibly reveal other significant interactions. The study’s findings are also limited by the specific characteristics of the sample population (resistance training for at least 1 year) and may not directly apply to other populations (e.g., sedentary adults, seniors, children, and others). In contrast, this population limitation might be viewed as an opportunity for future research to investigate these different perspectives (e.g., a wider range of populations). The findings are also specific to the constraints inherent in the study design, including the duration and intensity of the stretching protocols and the timing of measurements. Furthermore, while we utilized the best performance scores for analysis, others have suggested that the average scores may provide more consistent values [66,67]. Finally, for the DJ, although we observed and provided corrections, we could not be certain that there was no change in fall height and thus the DJ height may not have been consistently and exactly 20 cm for each participant. These limitations underscore the need for cautious interpretation and suggest paths for future research to enhance the validity and generalizability of findings in stretching interventions. Although the participants practiced the tests in the familiarization session, unilateral CMJ and DJ are movements many participants were unfamiliar with.

## 6. Conclusions

In summary, both an HISD and LILD SS intervention of the dominant leg hamstrings resulted in increased stretched leg ROM (non-significant (*p* = 0.06) for LILD stretched leg) with no significant changes in contralateral ROM and an overall (main effect for time) decrease in unilateral force, instantaneous strength, and associated EMG. It is possible that both interventions had insufficient influence on increased stretch tolerance for the contralateral limb. LILD induced a significant increase in the stretched leg unilateral DJ height.

## Figures and Tables

**Figure 1 sports-12-00257-f001:**
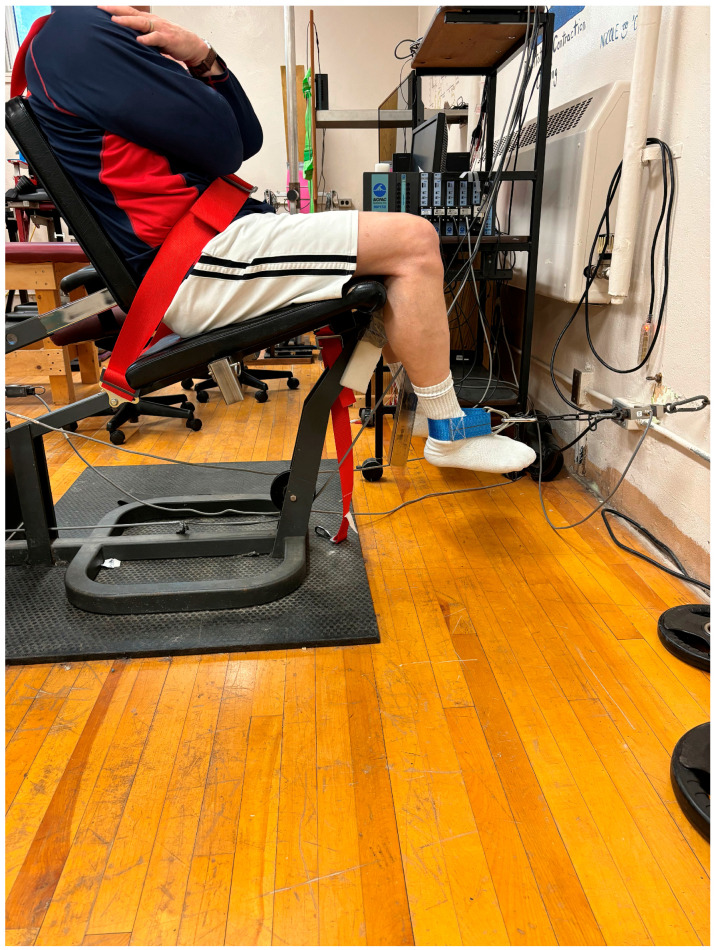
Isometric knee flexion apparatus.

**Table 1 sports-12-00257-t001:** Pre- and post-test data.

	Stretched Leg					
	HISD Pre	HISD Post	LILD Pre	LILD Post	Control Pre	Control Post
ROM (°)	106.0± 9.0# *p* = 0.008	110.6 ± 12.6# *p* = 0.008	109.3 ± 16.2 + *p* = 0.06	112.2 ± 16.5 + *p* = 0.06	107.2 ± 11.9	107.8 ± 11.8
MVIC (N)	350.6 ± 99.1	360.1 ± 101.6	349.0 ± 101.6	336.3 ± 114.9	354.6 ± 100.7	347.4 ± 108.7
MVIC EMG (mV)	0.083 ± 0.038	0.086 ± 0.039	0.079 ± 0.038	0.081 ± 0.039	0.086 ± 0.041	0.081 ± 0.037
IS (N)	126.1 ± 77.4	106.8 ± 75.4	116.5 ± 77.6	111.8 ± 72.5	119.2 ± 74.8	118.1 ± 83.5
IS EMG (mV)	0.059 ± 0.038	0.055 ± 0.033	0.059 ± 0.028	0.057 ± 0.038	0.064 ± 0.030	0.063 ± 0.037
CMJ (cm)	11.2 ± 5.2	11.1 ± 4.9	10.6 ± 3.7	10.6 ± 4.3	10.7 ± 4.4	10.6 ± 5.1
DJ (cm)	11.0 ± 4.4	10.7 ± 4.9	10.3 ± 4.4	10.5 ± 3.7 * *p* < 0.05	11.9 ± 4.2	11.1 ± 4.2
	**Contralateral Non-Stretched Leg**					
	**HISD Pre**	**HISD Post**	**LILD Pre**	**LILD Post**	**Control Pre**	**Control Post**
ROM (°)	104.2 ± 11.1	105.3 ± 14.2	108.5 ± 12.7	109.1 ± 15.3	105.8 ± 12.7	107.2 ± 13.3
MVIC (N)	344.0 ± 107.0	344.8 ± 96.5	336.7 ± 101.1	320.2 ± 103.2	343.2 ± 99.6	330.8 ± 94.7
MVIC EMG (mV)	0.076 ± 0.030	0.074 ± 0.028	0.069 ± 0.027	0.066 ± 0.028	0.073 ± 0.023	0.072 ± 0.029
IS	104.0 ± 83.6	112.9 ± 75.3	103.9 ± 74.9	113.1 ± 82.8	113.1 ± 82.8	108.6 ± 85.1
IS EMG (mV)	0.061 ± 0.059	0.051 ± 0.029	0.045 ± 0.025	0.045 ± 0.028	0.052 ± 0.032	0.049 ± 0.030
CMJ (cm)	9.9 ± 4.6	10.2 ± 5.1	11.1 ± 4.3	9.7 ± 3.9	10.0 ± 4.6	9.0 ± 4.1
DJ (cm)	9.4 ± 3.8	9.3 ± 3.9	10.8 ± 4.8	9.1 ± 3.8 * *p* < 0.05	9.4 ± 4.1	10.2 ± 5.2

CMJ: countermovement jump, DJ: drop jump, HISD: High intensity, low duration, LILD: Low intensity, high duration, IS: instantaneous strength (peak force exerted in the first 100 ms of the MVIC), MVIC: maximal voluntary isometric contraction, ROM: range of motion. The hashtag or number (#) symbol highlights a significant difference between HISD pre- and post-tests. Asterisks (*) represent a significant difference between the post-tests of the stretched and contralateral legs.

## Data Availability

Data are available upon request.

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
