# Peer review of "The Effects of High-Intensity, Short-Duration and Low-Intensity, Long-Duration Hamstrings Static Stretching on Contralateral Limb Performance"

_sports, 2024, doi:10.3390/sports12090257_

Round 1

Reviewer 1 Report

Comments and Suggestions for Authors

The improvements described in the introduction must be contextualized (sports or fitness fields).

The application should be more specific if only the acute effect is sought or the stable effect over time, since flexibility work has many applications, restoration, or development, among others.

In the participants, it should be indicated that in addition to strength training, they include specific flexibility conditioning during the week in order to be able to differentiate the starting point.

Is a 5' warm-up on a cycloergometer sufficient for the good development of the experiment?

A 5% increase in hamstrings may have been produced by prior learning. The lapse between them is very short.

Is the warm-up of the maximum isometric contraction after the hamstring work, or was it prior to isometric contraction?

How did you determine 50 or 90% of maximum contraction? Did you calculate 100% first? this is not clear.

It is logical to think that there is less activation with extensive work as there is more inhibition, but in the discussion, it is missing to indicate in greater depth the scope of application of this study, for example, to indicate in which contexts to use one or the other method depending on the objective.

Given that there was a sample of men and women, the differentiation of the results between the two would be very interesting and deserves a section to be commented on.

Author Response

See attached file for authors' response.

Reviewer 2 Report

Comments and Suggestions for Authors

General comments

This is a well written study with some nice points, however, there are a number of inaccuracies and errors that need to be acknowledged within the manuscript before it can be accepted for publication.

One consistent issue which is impacting readability is the use of abbreviations and non-standard abbreviations, this should be kept to a minimum and used consistently.

Specific comment

Title – As you are talking about duration or a period of time you should use the terms long and short duration. At one point you even clarify and identify this, so you should be consistent,

Abstract

L22 – You suggest magnitude, so you should provide an effect size.

L25 – You suggest “significantly” but the alpha error was p=0.05, this wouldn’t be significant.

Introduction

L52 – Large magnitude – provide the effect size.

L85-86 – Please ensure clarity on these assessments being performed unilaterally.

Methods

L89 – Great to see the a-priori sample size estimation, could you provide the effect sizes from referenced studies.

L102-104 – Please provide the declaration of Helsinki confirmation.

L124 – Figures of assessments would be useful for all assessments.

L154 – Why take the highest peak when you should take the average?

Henry, F. M. (1967). “Best” versus “Average” Individual Scores. Research Quarterly. American Association for Health, Physical Education and Recreation38(2), 317–320. https://doi.org/10.1080/10671188.1967.10613396

Whitley , J. D. and Smith , L. E. 1963 . Larger correlations obtained by using average rather than “best” strength scores . Res. Quart. , 34 : 248 – 49 .

L156 – How was onset calculated if calculating a measure of rapid force?

L173- How did you cue the CMJ and DJ? As this can effect importance.

L179-180 – I would avoid the use of terms eccentric and concentric, based of current reccomendations.

McMahon, John J., et al. "Understanding the key phases of the countermovement jump force-time curve." Strength & Conditioning Journal 40.4 (2018): 96-106.

Hahn D. On the phase definitions of counter movement jumps. Scand J Med Sci Sports. 2023 Mar;33(3):359-360. doi: 10.1111/sms.14288. PMID: 36775878.

L186 – Why 20cm? Can you calculate actual fall height? Currently based drop jump literature there is about 20% change in drop drop height and actual fall height (either +/-)

Geraldo, G.F.; Bredt, S.T.; Menzel, H.J.; Peixoto, G.H.; Carvalho, L.A.; Lima, F.V.; Soares, J.S.; Andrade, A.G. Drop height is influenced by box height but not by individual stature during drop jumps. J. Phys. Educ. 201930, e-3078.

Costley, L.; Wallace, E.; Johnston, M.; Kennedy, R. Reliability of bounce drop jump parameters within elite male rugby players. J. Sports Med. Phys. Fit. 201858, 1390–1397.

L212 – What about anything above 0.8? Currently it would read as if only 0.8 = large magnitude.

L213 – You need to report the pairwise effect sizes as they would provide a better understanding of the magnitude.

Table 1 – abbreviations should appear below the table in a final row, with only the title should be above.

L234 – EMG associated with instantaneous strength doesn’t make sense? This needs further explanation.

Discussion

L263-264 – This is where you use the more appropriate terms of longer and shorter duration, so be consistent through the manuscript using these terms.

L279 – You have already abbreviated DNIC, however, as you only use it twice there is no need to abbreviate. This would improve readability,

L296 – Suggesting some this is “near significance” is inappropriate, it either is or isn’t based of probability and you cannot be near significance.

Harvey, L.A. Nearly significant if only…. Spinal Cord 56, 1017 (2018). https://doi.org/10.1038/s41393-018-0214-8

L313-317 – Not sure on where this has come from, it doesn’t add anything to the manuscript.

L336 – What impulse increases be specific? As it relative net impulse which will dictate acceleration and thus take off velocity and jump height, but if the person the spending long on the ground with a more compliant strategy, with only a change in strategy rather than kinetics or force production.

L346 – Not necessarily improved based off increased sample size.

L344 – Is there anything you could do to overcome these limitations.

L357 – Condense the conclusion to a single paragraph.

L359 – See previous point on near significance.

L364-365 – How can you suggest long duration at high intensity as you have not included this within your study. Please stick to the findings of the present study.

L370 – Based off the methods used I would not suggest the improvements in DJ were desirable, more measures (such as strategy, outcome and kinetics) are needed to establish this.

Comments on the Quality of English Language

Please see specific comments

Round 2

Reviewer 2 Report

Comments and Suggestions for Authors

I appreciate and commend the authors on their changes made and believe the manuscript is much stronger for it. I have left it at minor amendments for a few reasons, there are a few minor changes I feel need to make, and some misunderstanding of previous comments.

I also want to continue some of the discourse raised in the review process, not necessarily that it impacts my opinion on the study. I think as scientists we need this appropriate discourse for our own learning and development.

I have highlighted the most recent comments.

“One consistent issue which is impacting readability is the use of abbreviations and nonstandard abbreviations, this should be kept to a minimum and used consistently.”

Authors’ response:

Most of our abbreviations or acronyms are commonly used such as ROM: range of motion, SS: static stretching, MVIC: maximum voluntary isometric contraction, CMJ: countermovement jump, DJ: drop jump

The intervention acronyms HILD and LIHD are used extensively throughout the manuscript and thus should be easily familiar to the reader. We removed one acronym from the abstract: POD. We have also removed MTU from the manuscript and replaced with the full term and noticed that POD was not defined so we have inserted the full term (point of discomfort) at each occurrence. MVIC was also not defined so it is now properly defined in its first appearance.

Reviewers’ response – I appreciate the change and readability is improved. It is worth considering sometimes if abbreviations or acronyms are worthwhile, even if very standard if impacts readability by using too many can have detrimental effects. Appreciate the changes made.

Specific comments

Abstract

L22 – You suggest magnitude, so you should provide an effect size.

Authors’ response: We have added the following detail re: effect size magnitudes as follows:

“Following the HISD condition, the post-test ROM for the stretched leg (110.6±12.60 ) exceeded the pre-test (106.0±9.00 ) by a small magnitude effect of 4.2% (p=0.008, d=0.42). With LILD, the stretched leg post-test (112.2±16.50 ) exceeded (p=0.06, d=0.18) the pre-test ROM (109.3±16.20 ) by a non-significant, trivial magnitude of 2.6%.”

Reviewers’ response – I like the inclusion of the effect sizes, the last part of the sentence which provides a % change needs the effect size clearing identifying (I think this is the 0.18 identified earlier. As 2.6% change is not a magnitude-based statistics and doesn’t tell you if it is trivial or small. So, maybe just reordering of the results here.

Methods

L102-104 – Please provide the declaration of Helsinki confirmation. Authors’ response: Declaration added as requested.

Reviewers’ response – Add the year to the declaration of Helsinki, this is minimum practice to identify for ethics.

L124 – Figures of assessments would be useful for all assessments. Authors’ response: The assessments for hamstrings (hip flexion) ROM, MVIC, CMJ and DJ are ubiquitous in the literature and reported in hundreds of articles if not possibly thousands. Hence, we respectfully would suggest that adding figures to illustrate these common tests is not necessary.

Reviewers’ response – I would contest this, if they were reported in the literature in “hundreds if not thousand” which I would suggest not the latter, you would have cited these methods within the literature. Additionally, if it helps improve replication why would you not want to add a picture or figure of testing procedures. I think really only “Knee Flexion Maximal Voluntary Isometric Contraction (MVIC) Force” requires a figure for appropriate understanding by the reader. As this method is not that common, based many of the systematic reviews on hamstring strength assessments.

L154 – Why take the highest peak when you should take the average?

Henry, F. M. (1967). “Best” versus “Average” Individual Scores. Research Quarterly. American Association for Health, Physical Education and Recreation, 38(2), 317–320. https://doi.org/10.1080/10671188.1967.10613396

Whitley , J. D. and Smith , L. E. 1963 . Larger correlations obtained by using average rather than “best” strength scores . Res. Quart. , 34 : 248 – 49 .

Authors’ response: We wanted to use and analyze the best performance. We have used this procedure in literally hundreds of publications from my lab (1993 – 2024). This procedure has not been previously questioned.

Reviewers’ response – This is a crucial point that I would suggest considering further, what if the best performance in an anomaly that never occurs again, this would provide a false positive result. If you read the cited papers above you could make justifications for either average or best, I would still argue to use the average due to biological variation and the potential of anomalous results, but you can highlight this as a limitation based of the papers cited above, suggesting potential for anomalous results to be included. This doesn’t require a change to the paper.

Additionally, I would strongly advise the authors not to use the comment “We have used this procedure in literally hundreds of publications from my lab (1993 – 2024). This procedure has not been previously questioned.” This is an extremely poor reaction to a scientific discourse, just because it has never been questioned previously doesn’t mean it shouldn’t be questioned. All of history has shown practises that has been performed thousands or millions of times are necessarily correct and need further interrogation.

L156 – How was onset calculated if calculating a measure of rapid force?

Authors’ response: The following information has been added. “Onset of instantaneous strength was determined by visual inspection of the force output where the force line initially deviated from the baseline without returning to the baseline and continued to escalate.”

Reviewers’ response – This is good information, could be worth further investigation (not for the present study) to determine the reliability and validity of this method.

L173- How did you cue the CMJ and DJ? As this can effect importance.

Authors’ response: We have added the following information: “Participants initiated the jump at their personal discretion (i.e., when they felt ready).”

Reviewers’ response – I think you have misunderstood the comment here, cueing refers to “jump high as possible”, “jump as fast as possible” or a combination, as this can impact the jump performance and strategy. If you could explain this that would provide context.

L179-180 – I would avoid the use of terms eccentric and concentric, based of current recommendations.

McMahon, John J., et al. "Understanding the key phases of the countermovement jump force-time curve." Strength & Conditioning Journal 40.4 (2018): 96-106.

Hahn D. On the phase definitions of counter movement jumps. Scand J Med Sci Sports. 2023 Mar;33(3):359-360. doi: 10.1111/sms.14288. PMID: 36775878.

Authors’ response: The terms and eccentric and concentric have been removed as suggested.

Reviewers’ response – Line 324 this has not been removed, please check.

L186 – Why 20cm? Can you calculate actual fall height?

Currently based drop jump literature there is about 20% change in drop drop height and actual fall height (either +/-)

Geraldo, G.F.; Bredt, S.T.; Menzel, H.J.; Peixoto, G.H.; Carvalho, L.A.; Lima, F.V.; Soares, J.S.; Andrade, A.G. Drop height is influenced by box height but not by individual stature during drop jumps. J. Phys. Educ. 2019, 30, e-3078.

Costley, L.; Wallace, E.; Johnston, M.; Kennedy, R. Reliability of bounce drop jump parameters within elite male rugby players. J. Sports Med. Phys. Fit. 2018, 58, 1390–1397.

Authors’ response: No, we did not calculate actual fall height. As mentioned, we asked participants to step off the platform. We have added the following detail: DJ height trials commenced by participants standing on a platform 20cm high and stepping off (no prior vertical jump component) onto the tested leg. We can assure the reviewer that instructions were followed, and the jump procedures was consistent and reliable.

Reviewers’ response – I appreciate the addition, but unless you have 3D motion or calculate fall height from touch down velocity you cannot be certain that there was no change in fall height. Again, add this as a limitation that fall height might not have been controlled. If you read the studies cited above they asked participants to step/drop off and they still had the change in fall height of +/- 20%.

I would advise the authors to note that review process is to learn and develop, I’m not here to pull your study apart I want to improve it with future studies for yourself and others who may want to replicate your study. Any of these methodological limitations are purely that, limitations which you can then discuss in the limitations of the study. If I thought the limitations were large enough where the results were not accurate, misleading or scientifically incorrect, I would have rejected the study, as I said in my first review this study has merit.

Discussion

L313-317 – Not sure on where this has come from, it doesn’t add anything to the manuscript.

Authors’ response: We would respectfully disagree. This paragraph outlines possible neural, morphological, and psychological mechanisms underlying non-local decreases in force/strength. Mental energy deficits and increases in effort perception following an exercise bout are both mechanisms discussed in the literature.

Reviewers’ response – I understand where the authors are coming from, but I personally don’t think after static stretching there would be underlying psychological mechanisms related to the change, I still think this comes a little bit out of nowhere, but I am no sport psychologist, so I am happy to include if the authors think appropriate.

L344 – Is there anything you could do to overcome these limitations.

Authors’ response: Not really. As this was a student thesis, not all variables can be answered. It would be great to have an experiment that included resistance trained and non-resistance trained, youth, young adult, middle aged, and senior adults who attended a myriad of sessions that examined a multitude of stretching durations and post-stretching testing times but of course this is not practical. Hence, more studies are needed to attack these different perspectives.

Reviewers’ response – This is another clear point to make, that I don’t think not having these populations is a limitation. It is as you say “more studies are needed to attack these different perspectives”, it is a recommendation for future research. You can use the limitations highlighted above in this section and then provide support that further research is required in wider populations.

Author Response

Pleas ethank the reviewers for their work at improving the manuscript. Our responses to reviewer #2 second round are attached.
